# A gender-sensitised weight-loss and healthy living program for men with overweight and obesity in Australian Football League settings (Aussie-FIT): A pilot randomised controlled trial

Dominika Kwasnicka[1,2,3¤], Nikos Ntoumanis[1,2], Kate Hunt[1,4,5], Cindy M. Gray[5], Robert U. Newton[6], Daniel F. Gucciardi[1,7], Cecilie Thøgersen-Ntoumani[1,2], Jenny L. Olson[1,2], Joanne McVeigh[1,8,9], Deborah A. Kerr[10], Sally Wyke[5], Philip J. Morgan[11], Suzanne Robinson[10], Marshall Makate[10], Eleanor Quested[1,2]*

1 Physical Activity and Well-being Research Group, Curtin University, Perth, Australia, 2 School of Psychology, Curtin University, Perth, Australia, 3 SWPS University of Social Sciences and Humanities, Wroclaw, Poland, 4 Faculty of Health Sciences and Sport, University of Stirling, Stirling, United Kingdom, 5 Institute of Health and Wellbeing, University of Glasgow, Glasgow, United Kingdom, 6 Exercise Medicine Research Institute, Edith Cowan University, Perth, Australia, 7 School of Physiotherapy and Exercise Science, Curtin University, Perth, Australia, 8 School of Occupational Therapy, Speech Therapy & Social Work, Curtin University, Perth, Australia, 9 Movement Physiology Laboratory, School of Physiology, University of Witwatersrand, Johannesburg, South Africa, 10 School of Public Health, Curtin University, Perth, Australia, 11 Priority Research Centre in Physical Activity and Nutrition, School of Education, University of Newcastle, Newcastle, Australia

¤ Current address: NHMRC CRE in Digital Technology to Transform Chronic Disease Outcomes, University of Melbourne
* eleanor.quested@curtin.edu.au

## Abstract

### Background

Recent evidence shows that sport settings can act as a powerful draw to engage men in weight loss. The primary objective of this pilot study was to test the feasibility of delivering and to evaluate preliminary efficacy of Aussie-FIT, a weight-loss program for men with overweight/obesity delivered in Australian Football League (AFL) settings, in preparation for a future definitive trial.

### Methods and findings

This 6-month pilot trial took place in Perth, Australia. Participants were overweight/obese (Body Mass Index [BMI] $\geq$ 28 kg/m$^2$), middle-aged (35–65 years old) men. Participants were recruited in May 2018, and the intervention took place between June and December 2018. The intervention involved 12 weekly 90-min face-to-face sessions, incorporating physical activity, nutrition, and behaviour change information and practical activities delivered by coaches at 2 clubs. Data were collected at baseline and immediately postintervention. For trial feasibility purposes, 6-month follow-ups were completed. Outcomes were differences in weight loss (primary outcome) and recruitment and retention rates, self-

**Data Availability Statement:** The data are not freely available, as we did not have participant

consent to store de-identified data in an online depository. Enquiries can be directed to the Curtin University Human Research Ethics Committee (hrec@curtin.edu.au).

**Funding:** Aussie-FIT was funded by Healthway the Western Australian Health Promotion Foundation (EQ; grant number 31953): https://www.healthway. wa.gov.au/. The funders had no role in study design, data collection and analysis, decision to publish, or preparation of the manuscript. Open access of this article was financed by the Ministry of Science and Higher Education in Poland under the 2019-2022 program "Regional Initiative of Excellence", project number 012 / RID / 2018/19.

**Competing interests:** The authors have declared that no competing interests exist.

**Abbreviations:** AFL, Australian Football League; Aussie-FIT, Australian Fans in Training; BMI, Body Mass Index; CONSORT, Consolidated Standards of Reporting Trials; CSO, Chief Scientist Office; EQ-5D-5L, EuroQol five-dimensional five-level; FFIT, Football Fans in Training; GP, General Practitioner; IBQ, Interpersonal Behaviours Questionnaire; ICER, incremental cost-effectiveness; MRC, Medical Research Council; MVPA, moderate-to-vigorous physical activity; NIHR, National Institute for Health Research; PANAS, Positive and Negative Affect Scale; PSQI, Pittsburgh Sleep Quality Index; QALY, Quality-Adjusted Life Years; RCT, Randomised controlled trial; SDT, Self-Determination Theory; SPFLT, Scottish Professional Football League Trust; SRBAI, Self-Report Behavioural Automaticity Index; TIDieR, Template for Intervention Description and Replication.

reported measures (for example, psychological well-being), device-measured physical activity, waist size, and blood pressure at 3 months. Within 3 days of advertising at each club, 426 men registered interest; 306 (72%) were eligible. Men were selected on a first-come first-served basis ($n = 130$; $M$ age = 45.8, $SD = 8$; $M$ BMI = 34.48 kg/m$^2$, $SD = 4.87$) and randomised by a blinded researcher. Trial retention was 86% and 63% at 3- and 6-month follow-ups (respectively). No adverse events were reported. At 3 months, mean difference in weight between groups, adjusted for baseline weight and group, was 3.3 kg (95% CI 1.9, 4.8) in favour of the intervention group ($p < 0.001$). The intervention group's moderate-to-vigorous physical activity (MVPA) was higher than the control group by 8.54 min/day (95% CI 1.37, 15.71, $p = 0.02$). MVPA among men attracted to Aussie-FIT was high at baseline (intervention arm 35.61 min/day, control arm 38.38 min/day), which may have limited the scope for improvement.

## Conclusions

Aussie-FIT was feasible to deliver; participants increased physical activity, decreased weight, and reported improvements in other outcomes. Issues with retention were a limitation of this trial. In a future, fully powered randomised controlled trial (RCT), retention could be improved by conducting assessments outside of holiday seasons.

## Trial registration

Australian New Zealand Clinical Trials Registry: ACTRN12617000515392.

## Author summary

### Why was this study done?

- The prevalence of overweight and obesity is higher in men than in women in Australia (71% versus 56%).

- Provision of weight-loss programs suitable for men is limited, and men are less likely than women to take part in weight-loss programs.

- Use of sport settings, such as Australian Football League (AFL) club facilities, can be used as a powerful way to encourage men to participate in weight-loss programs.

### What did the researchers do and find?

- 130 men took part in the Aussie-FIT program. This men-only weight-loss program included 12 weekly 90-min sessions that included weight-loss education and coach-led physical activity. The AFL themed program was delivered in 2 Australian football club settings.

- After all of the men's initial weight and health measures were taken, half of the group received the program, and the other half waited for 3 months to receive it. At 3 months, the measurements were repeated so that we could compare differences in weight (and

other indicators of health) between men who had already completed the program and the men who were waiting to receive the program.

- We found that the men who had participated in the program lost on average 3.33 kg more than the men who had not. Men who had completed Aussie-FIT did, on average, 8.54 min more moderate-to vigorous physical activity per day than their counterparts who had not yet received the program.

## What do these findings mean?

- Aussie-FIT is feasible to deliver in Australian football settings. This study provides some initial evidence that Aussie-FIT may be appealing and effective as a weight-loss program for men with overweight and obesity in Australia.

- This study was relatively small (130 men took part). In order to make definitive policy recommendations regarding program scale out, a follow-up study involving more participants and more football clubs is needed. Also, studies with longer follow-up are needed.

- A limitation of this study was that the program did not attract men from culturally or linguistically diverse backgrounds or indigenous Australians; future studies should explore how the program can be made more appealing and acceptable to men from more diverse backgrounds.

## Introduction

Obesity is a global public health issue, and prevalence is increasing; in 2016, over 1.9 billion people worldwide were overweight, and over 650 million were obese [1]. In Australia in 2016, approximately 60% of adults were estimated to be overweight, including 25% classified as obese [2]. Although in Australia, overweight and obesity are more prevalent in men than in women (71% versus 56%) [3], men are underrepresented in trials of weight loss, and current community programs have not appealed to them [4,5]. The prevalence of overweight and obesity in men and low participation in trials may be in part because cultural constructions of masculinity in Western societies often promote overeating and excessive alcohol consumption and stereotype dieting or healthy eating as a 'female-only' activity [6,7,8]. Weight-loss programs are typically marketed to females and are rarely tailored to attract and meet the needs of men [6,9].

Professional sports can be used as a powerful draw to attract men to weight-loss programs [10,11]. For example, the Football Fans in Training (FFIT) program was designed to appeal to soccer fans with overweight and obesity in Scotland and to support them in losing weight, by fostering changes in their dietary and physical activity behaviours [12,13,14]. The program was delivered over 12 weekly sessions in the context of professional soccer (for example, premier league stadia, training facilities) and was effective in reducing weight (baseline-adjusted mean difference in weight loss between intervention and control groups at 12-month follow-up was 4.94 kg). FFIT also improved health behaviours and psychological well-being [14], and these effects were maintained 3.5 years postbaseline [15].

FFIT has been successfully adapted for other professional sport settings and for different countries, including rugby in England [16] and New Zealand (Rugby Fans in Training

[RuFIT] [17]) and hockey in Canada (Hockey FIT [18] and HAT TRICK [19]). The model has also been successfully used to inform healthy lifestyle programs for other outcomes, as in Euro-FIT, which focussed on physical activity and sedentary time, delivered in 4 European countries [20]. To date, this program has not been evaluated in Australia, and programs that address unique masculinity constructs are still an exception in Australia [21].

Building on FFIT, we developed the Australian Fans in Training (Aussie-FIT) program, a culturally sensitised version of FFIT for men in Australia, for delivery in Australian Football League (AFL) settings. The key features of Aussie-FIT have been described elsewhere [22]. Key innovations included the program being further developed in the context of behaviour change theories summarised in a recent theory review [23] and on reviews of successful studies of weight-loss maintenance [24]. This included the integration of principles from Self-Determination Theory (SDT) as a means to strengthen the quality, quantity, and longevity of motivation to change health behaviours [25]. According to SDT, humans have 3 psychological needs. These include the need to feel that one is competent (i.e., has the skills, support, and resources to meet challenges or strive for goals), autonomous (i.e., that what one is doing reflects personal choice, values, and volition), and related to others (i.e., respected, supported, cared for, and connected). A plethora of studies have indicated health behaviour change to be more effective and well-being to be enhanced when these needs are satisfied. Need satisfaction is important to foster because it almost unequivocally predicts more autonomous (i.e., self-determined, valued, and relevant) motives [25] which, according to a recent systematic review [26], are strongly related to long-term weight maintenance and physical activity behaviour. Accordingly, principles of SDT were applied throughout the program via integration into coach training and program activities. In line with theoretical understanding [23] and previous studies [24], men need to effectively self-regulate their behaviour and successfully form healthy habits to maintain long-term behaviour change. Social support and environmental factors play a crucial role in health behaviour change and long-term maintenance [9,21]. Men need access to plentiful resources to maintain their behaviour long-term, including physical (access to healthy food, exercise equipment), psychological (energy levels, positive mood), and social resources to facilitate health behaviour change maintenance [27,28].

The overarching aim of the study was to test the feasibility of delivering and evaluating the 12-week Aussie-FIT program and to test preliminary efficacy in the context of AFL in Australia to determine the appropriateness of a future randomised controlled trial (RCT). Specifically, the primary objectives were to examine (1) the feasibility of recruiting the target sample and retaining them at 3- and 6-month follow-ups; (2) the appropriateness of the selected measures to examine effects and to report effects of the intervention at 3-month follow-up on body weight, waist circumference, blood pressure, device-measured time spent in moderate-to-vigorous physical activity (MVPA) and sedentary time, self-reported dietary screener, and a range of secondary psychological outcomes; (3) the feasibility of administering the data collection protocols and questionnaire administration procedures in the AFL club setting at all 3 time points; and (4) the feasibility of collecting data to develop and undertake preliminary tests of a model to determine the cost-effectiveness of the program. Additional feasibility outcomes will be examined in a full process evaluation, reported elsewhere.

## Methods

### Study design

The Curtin University Human Research Ethics Committee (HREC2018-0458) provided ethical approval; all participants provided written informed consent before participation. The study protocol has been published elsewhere [22]. We undertook a two-group pilot waitlist

RCT from April–December 2018 in the Perth metropolitan area of Western Australia. The waitlist control treatment was chosen to align with FFIT RCT design and to minimise any ethical concerns related to withholding treatment in the control group. Participating groups were stratified by site (2), and measures were collected at baseline and 3 and 6 months postbaseline. The waitlist control group started participation in the program at 3 months; therefore, the 6-month measure represents a 'postprogram' assessment for the control arm and a short-term maintenance effect for the intervention arm. The Aussie-FIT pilot study was designed to inform the development of a definitive future trial. The study sample size followed guidelines for pilot RCTs [29,30].

## Participants

Men were recruited through a variety of methods, including features in weekly fan emails from the participating AFL clubs, social media announcements from the clubs and directly from the Aussie-FIT project social media accounts, word of mouth, and via the Aussie-FIT website [31]. We invited potential participants to register their interest and complete an eligibility check by visiting the program website and providing their age, weight, height, contact details, and availability to attend Aussie-FIT sessions. The study inclusion criteria were men aged 35–65 years, with Body Mass Index (BMI) 28 kg/m$^2$ or higher (eligibility based on baseline measures). Age and BMI cutoffs were based on the FFIT study [32]. The study exclusion criteria were men who were unable to comprehend English, men who were unable to provide informed consent and/or agree to the randomisation, and those who were already participating in a health promotion or weight-loss program.

## Randomisation and blinding

The randomisation sequence was generated by a researcher not involved in the trial using SPSS with block sizes of 4, stratified by site and BMI category (with 4 BMI categories: <30, 30–34.9, 35–39.9, and >40 kg/m$^2$), and concealed until conditions were assigned. After baseline measures were taken, participants were allocated to the intervention or waitlist control group according to the randomisation sequence and informed about their group allocation via email and phone call. At follow-up measurements, we asked participants not to disclose their group allocation to research assistants; the latter were trained to avoid engaging in extensive conversations with the men during the weight assessment. The weight assessments were taken in a separate area for participant privacy and also to help mask assessors from hearing men mention their experiences of the program at follow-ups. An independent researcher blinded to the study allocation analysed study data. The study is reported in line with Consolidated Standards of Reporting Trials (CONSORT) guidelines extension for randomised pilot and feasibility trials [33] and the Template for Intervention Description and Replication (TIDieR) checklist [34] (S1 Table).

## Procedure

Full details of the procedure, intervention content, and measurement tools are provided in the study protocol [22]. Before delivery of the Aussie-FIT program, the research team received feedback on the program handbook from a purposive sample of consumers (i.e., male AFL fans; $N = 5$). Small adjustments were subsequently made based on this feedback (for example, some content was reworded or reordered). Six coaches (4 male and 2 female) with experience of coaching in community or recreational programs were recruited to deliver the program. The intention was for all coaches to be recruited directly from the participating clubs. In reality, 3 coaches were recruited via the AFL clubs, and 3 additional coaches were identified by the

research team because of limited capacity within the clubs to provide additional coaches. The Aussie-FIT coaches had experience in coaching and also professional experience in teaching, exercise instruction, or sport science. The coaches attended 4 half-day program delivery training workshops. These were delivered by 2 female members of the research team (EQ, DK) who had expertise in health psychology and motivation and behaviour change principles and experience designing and delivering coach education workshops (EQ). The program was delivered at the 2 AFL clubs in Perth, Western Australia. There were 2 deliveries to groups of approximately 15 men at each club (on 2 different days of the week) in each group (i.e., intervention and control). In total, there were 4 groups of men in the intervention arm, and 4 groups of men in the waitlist control group received the intervention at a later date.

Aligned with the FFIT program, the Aussie-FIT intervention included 12 weekly 90-min sessions designed to promote physical activity, healthy eating, and weight loss. The program was delivered to groups of approximately 15 men by one coach and included classroom-based activities and physical activity sessions. In the early weeks, a little more than half of the session was dedicated to classroom activities, with less time allotted to the physical activity sessions. Over the course of 12 weeks, the balance shifted to expand the physical activity component to align with the men's progress in fitness. The delivery style was informal, encouraging positive social interaction, humour, and 'friendly banter'. The program was gender-sensitised with an emphasis on dietary and physical activity changes that are consistent with masculinised practices (for example, ordering salad with a steak, getting back to being able to play football with children or grandchildren, discussion of the role of alcohol in weight loss, and fostering group support). The program supported participants to make small sustainable changes to their eating through portion control; reduced consumption of sugary drinks, energy-dense foods, and alcohol; and a gradual increase in physical activity by choosing the activity that the men enjoy the most or could most easily incorporate into daily life. To make the program culturally appropriate, program content was adapted to reflect the Australian Guidelines for Healthy Eating [35], and resources from 'LiveLighter' (a healthy eating campaign in Australia) were built into the program to illustrate key principles (for example, a tool for ease of reading food labels). The program was designed to teach participants strategies for self-regulation, goal setting, and avoiding compensatory behaviours (for example, overeating after intense physical activity) and to prevent relapse.

Building on the FFIT program [14], Aussie-FIT added new content to both the coach training and intervention content. In particular, new coach training included information on what coaches can say and do to create more need supportive and less controlling environments and why. This innovation was added because need support is known to promote self-determined motivation for behaviour change [36], and increased focus on this aspect of the intervention could further contribute to behaviour change and maintenance. This content was delivered in coach training with discussion of the basic principles of the theory, interactive activities (for example, scenarios, role-playing), detailed descriptions of these environmental components, and specific planning activities in which coaches detailed in writing what they planned to do during each session to be need supportive and how.

The program also included specific activities in which participants reflected on their personal motives and identified sources of basic psychological need satisfaction. In-session activities were included that encouraged men to recognise their own experiences of need satisfaction and sources of autonomous motivation. Previous studies reported that focus on weight-loss maintenance should be incorporated from the beginning of the weight-loss program [23,37,38]; therefore, we placed greater emphasis from the start on relapse prevention and on long-term maintenance of behavioural changes. Building on the content of the original FFIT program, Aussie-FIT participants were supported in how to best form habits [14] and

how to form specific action and coping plans (expanding on their initial SMART goals); these plans were revisited and revised during subsequent Aussie-FIT sessions.

Other innovative aspects of the Aussie-FIT intervention included participants and coaches being invited to join closed Facebook groups which comprised the approximately 15 men in their Aussie-FIT group and their coach. Automated text messages, written in language to promote feelings of autonomy, competence, and relatedness, were sent each week to encourage session attendance and included a brief description of the topic of the upcoming session (for example, "*Today we will talk about junk food—it can be tough to cut it out altogether, but there are plenty of ways you can make it healthier*"). In session 1, participants received an Aussie-FIT booklet with session summaries and space to complete in-session activities and to self-monitor their weight-loss progress and goals. Men also received activity monitors (Fitbit Zip), club t-shirts, and reusable 'LiveLighter' branded water bottles. Additional session information (for example, summaries of key points covered in the program) was available online, via a password-protected subsection of the program website.

There was no formally structured intervention content or contact provided after 12 weeks. Participants were free to communicate through the Facebook group with the coach and with each other. We encouraged the participants to keep in touch and to schedule to meet and to exercise together after the program had finished.

## Data collection

Measures were collected at the 2 AFL club facilities where we also delivered the Aussie-FIT sessions (alternative arrangements were made for men who could not attend measurement sessions at the club [for example, meetings at university premises]). After a brief welcome and summary of what the assessment session involved, men provided informed consent and completed measures at supervised stations in the following order: (1) weight, height, and waist circumference; (2) questionnaires completed on iPads; (3) blood pressure and health screening; (4) ActiGraph fitted and instructions given for use; and (5) final questions answered. As a thank-you gesture, men received an AUD$20 team store voucher at the completion of each assessment session.

During health screening, men completed the Adult Pre-exercise Screening System form [39] to identify contraindications to participate in the physical activity sessions. Following the recommendations of Exercise and Sport Science Australia [40], participants whose responses on the screening form or blood pressure readings raised a concern had the opportunity to discuss with an allied health professional (i.e., university-qualified practitioner with expertise in preventing, diagnosing, and treating a range of conditions and illnesses) how to adapt the physical activity components of the program to meet their needs. Thirteen participants with a systolic blood pressure reading of over 160 mm/Hg were advised to seek an opinion from their General Practitioner (GP) prior to commencing the program; none were considered unfit to participate by their GP.

## Measures

The primary outcome in this trial was mean difference in weight between groups at 3 months, adjusted for baseline weight. Secondary outcome measures were feasibility of recruitment and trial retention; device-measured weight, waist, and blood pressure; physical activity; self-reported diet and alcohol; well-being; quality of life; motivation for physical activity and other variables relevant to behaviour change; and self-reported sleep.

**Feasibility of recruitment and trial retention.** Recruitment rates were recorded as the number of men who registered interest in the trial either via a web-based form or over the

phone. The online form included a self-administered screening tool to check men's age and BMI met the inclusion criteria. We also recorded the number of men who were retained to the randomisation stage and those who attended follow-ups at 3 and 6 months. Feasibility-related outcomes are reported descriptively.

**Device-measured weight, waist, and blood pressure.**    Weight was assessed using an electronic scale (Seca 813 Flat Scale, EMSE81; Birmingham, United Kingdom). Waist circumference was measured twice, or 3 times if 2 measures differed by more than 5 mm; at each time point, we took the average of these measures. Resting blood pressure was measured with a digital blood pressure monitor (Omron HEM-705CP; Milton Keynes, United Kingdom) after sitting for at least 5 min. If systolic blood pressure was over 139 mm/Hg and/or diastolic blood pressure was over 89 mm/Hg, 2 further measures were taken and recorded, and a mean was calculated from the second and third measures.

**Physical activity.**    To provide a device-based measure of physical activity and sedentary behaviour/time; participants were fitted with a hip-worn ActiGraph GTX-9 (ActiGraph, Pensacola, FL, USA) [41] accelerometer that they were asked to wear continuously (24 h/day) for the following 9 days. The GTX-9 was programmed to record raw data at a frequency of 30 Hz, which were later reduced to vertical axis movement counts of 60-s epoch for the purpose of the current analyses. Participants were instructed to wear the accelerometer on the right hip continuously except during bathing or aquatic activities. Accelerometer data were downloaded using ActiLife version 6.5.4, and 1-min epoch data were processed using a validated algorithm in SAS (version 9.3) [42]. Common cut-points [43,44] were used to classify each minute as sedentary (<100 counts per minute, cpm), light intensity (100–1,951 cpm), moderate intensity (1,952–5,724 cpm), or vigorous intensity (>5,724 cpm). The uncensored step count (i.e., all steps counted) recorded per minute was also used. All participants with ≥4 valid days of data were included in the analyses of physical activity and sedentary behaviour.

**Self-reported dietary screener and alcohol assessment.**    Nutrition relevant outcomes were assessed using the Dietary Instrument for Nutrition Education-based measures [45] adapted for the Australian population [46]. Briefly, the dietary screener asked participants to report how many times over the past 7 days they ate or drank specific foods. From the responses, we calculated the average change in fatty food score, fruit and vegetable score, sugary food score (all 3 on a 1–4 scale, with higher scores indicative of higher consumption). We used a 7-day recall method to measure alcohol consumption, and based on the calendar recall data, we calculated total alcohol consumption (reported as total average number of alcohol units consumed in a week; 1 unit is 10 ml of pure alcohol).

**Well-being, quality of life, and motivation-related variables.**    We used the 10-item Rosenberg self-esteem scale (range 1–4), for which higher scores indicate higher self-esteem [47], and the 10-item Short Form of the Positive and Negative Affect Scale (PANAS) separating the 2 constructs [48] (range 1–5 for both), for which higher scores indicate higher positive or negative affect. The 12-item Interpersonal Behaviours Questionnaire (IBQ) was applied to assess men's perceptions of the psychological need support in relation to weight loss that they received from family and friends (range 1–7), with a higher score indicating more support [49]. We used items from an adapted measure of the Treatment Self-Regulation Questionnaire of weight-loss motivation [50] to assess autonomous and controlled motivation regulating weight-loss behaviours (range 1–5), where higher scores indicate higher autonomous motivation [50]. We assessed autonomy, competence [51], and relatedness [52] psychological need satisfaction in relation to weight-loss behaviours and collapsed all needs into 1 score for the purposes of the analysis (range 1–5), with higher scores indicative of higher need satisfaction. We measured health-related quality of life using Quality-Adjusted Life Years (QALYs) derived from the EuroQol five-dimensional five-level version (EQ-5D-5L) [53] and EQ-VAS overall

score (range 0–100), for which high scores indicate better overall health. We converted the EQ-5D-5L values into utility weights on the basis of the preference weights of a pilot sample from the Australian general population to calculate QALYs [54].

**Variables relevant to behaviour change.** We also assessed goal facilitation and competing goals in relation to weight-loss goals [55], barriers [56] and planning [57] (all scores range 1–5), and habits using the Self-Report Behavioural Automaticity Index (SRBAI) for physical activity and for healthy eating [58] (both scales range 1–7). For all these measures, higher scores are indicative of improvement on the construct assessed.

**Self-reported sleep.** We used the Pittsburgh Sleep Quality Index (PSQI) to assess 7 sleep components; a score of zero indicates the highest possible quality sleep, and 21 indicates the worst sleep quality [59].

### Statistical and economic analyses

Statistical analyses included descriptive statistics and percentages for each group. We used a one-way random effects analysis of covariance in Mplus 8.2 [60] to estimate the treatment effect between groups [61,62], adjusting for baseline values of the dependent variables and clustering effects (teams of 15 men). We did not stratify by club because we only had 2 clubs participating, both based in Perth, Western Australia. We conducted an all-cases analysis that involved all participants who were randomised to the intervention or control group and provided at least baseline data; missing data were handled using full information maximum likelihood estimation [63]. We compared the baseline characteristics of the participants who dropped out with the characteristics of the participants who completed the full study to check for differences between the 2 groups. Secondary analyses focussed on maintenance effects for the intervention group and treatment effects for the control group; these analyses were conducted on each group separately. In separate, within-group analyses, we modelled the linear effect of time (3 to 6 months) as a predictor of dependent variables, adjusting for baseline values and the clustering effect of groups. In so doing, the effect can be interpreted as the average amount of change on the dependent variable between 3 and 6 months.

Economic analysis included developing and piloting an economic model to test procedures and measures that could be used to estimate the cost-effectiveness of the program in a future full-scale trial. Direct costs associated with the program included program set-up, promotion, and delivery material costs. As a gauge of preliminary cost-effectiveness, the cost per 5% weight reduction and cost per QALY were assessed using the EQ-5D-5L data. The values of these data were converted into utility weights on the basis of the preference weights of a pilot sample from the Australian general population [54].

## Results

### Recruitment and retention

We assessed 426 men who applied online to participate in the program for eligibility; 120 did not meet the inclusion criteria (Fig 1). Places were allocated on a first-come, first-served basis; 130 men were invited to undertake baseline measures, all of whom were assessed and considered eligible and randomised to the intervention ($n = 64$) or waitlist control group ($n = 66$). One hundred and seventy-six were not offered a place in the program because of limited capacity in this study. Study retention was moderate, with 50 participants (78%) completing 3-month follow-up measures in the intervention group and 62 (93%) in the waitlist control group. For 6-month measures, 35 participants (54%) in the intervention group and 47 participants (71%) in the control group were retained. Assessment of program attendance was not feasible because not all coaches followed the protocol and regularly checked attendance.

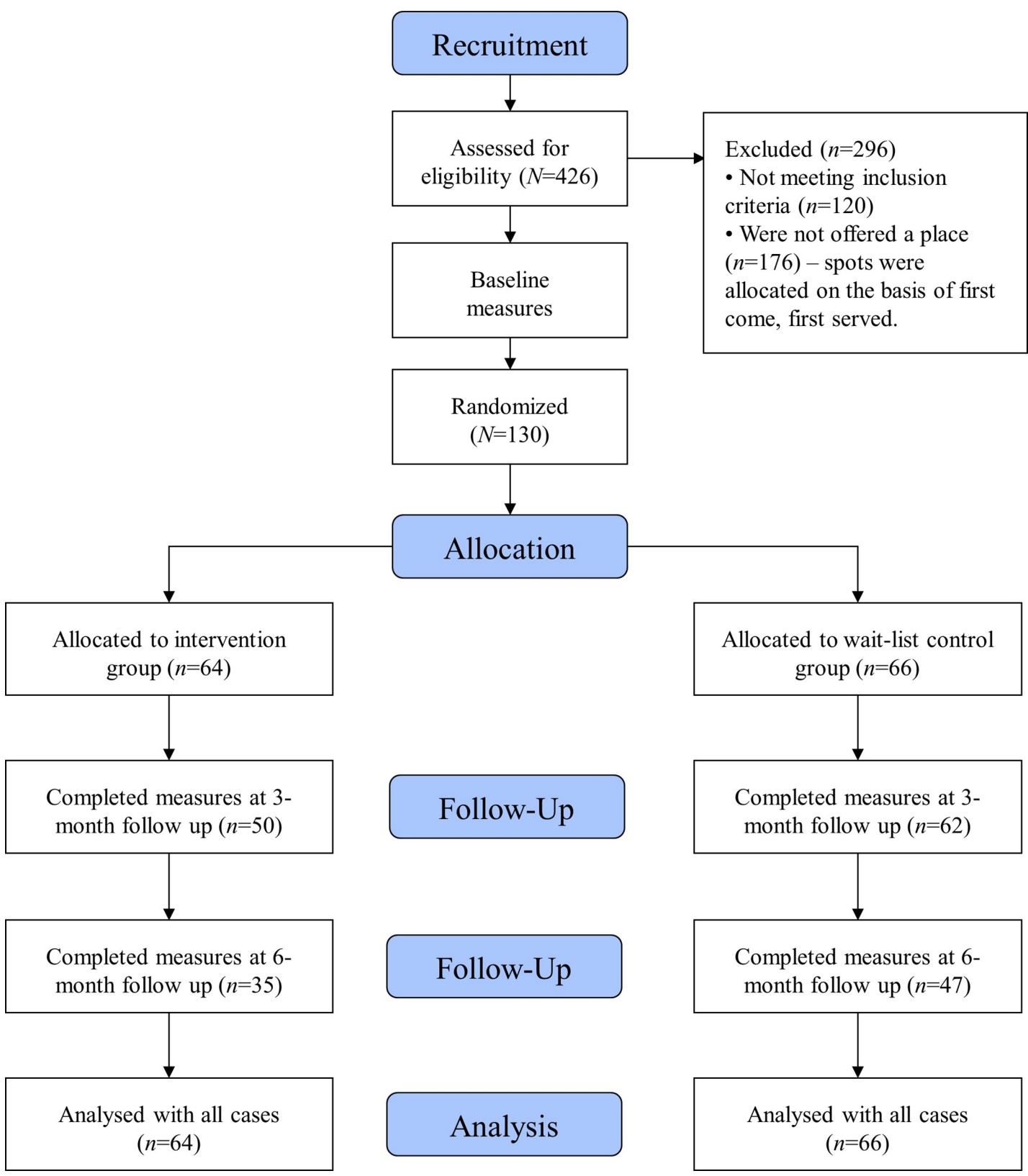

**Fig 1. CONSORT flow diagram for the Aussie-FIT pilot trial.** Aussie-FIT, Australian Fans in Training; CONSORT, Consolidated Standards of Reporting Trials.

## Sample characteristics

Demographics, device-derived outcome measures, and scores for self-reported outcomes for the intervention, control group, and total sample at baseline are provided in Table 1. Accelerometer data at all 3 time points for participants with 4+ days of valid (i.e., $\geq$10 h/day) wear time are presented in S2 Table. At baseline, 22 men were categorised as overweight and 108 men as obese (83% of total sample), with a mean weight of 111.4 kg ($SD$ = 18.2) and mean waist circumference of 116.2 cm ($SD$ = 13.7). Mean baseline blood pressure (137.8 mm/Hg [$SD$ = 16.97] systolic and 87.3 mm/Hg [$SD$ = 9.8] diastolic) was at the higher end of recommended levels. At baseline, the mean number of steps per day was high (10,928 [$SD$ = 3,454]), with 35.3 ($SD$ = 19.2) min of MVPA per day (uncensored, unless otherwise specified).

## Preliminary analysis

There were no significant differences in any baseline characteristics of the participants who dropped out compared to those who completed the study; all scales were reliable at all 3 time points (using Cronbach's alpha to measure internal consistency; S1 Text).

**Table 1. Baseline characteristics of participants allocated to the intervention arm or waitlist control arm.**

|  | Aussie-FIT Intervention Group ($n$ = 64) | Aussie-FIT Waitlist Control Group ($n$ = 66) | Total ($N$ = 130) |
|---|---|---|---|
| Age (average years, SD) | 44.2 (7.6) | 47.2 (8) | 45.8 (7.9) |
| Ethnic origin (%) |  |  |  |
| —White | 61 (95) | 63 (95) | 124 (95) |
| —Mixed | 2 (3) | 2 (3) | 4 (3) |
| —Other | 1 (2) | 1 (2) | 2 (2) |
| Employment status (%) |  |  |  |
| —In paid employment/self-employed | 60 (94) | 61 (92) | 121 (93) |
| —Retired | 2 (3) | 0 (0) | 2 2 (2) |
| —Other | 2 (3) | 5 (8) | 7 (5) |
| Average number of hours worked per week (SD) | 42.01 (11.74) | 40.16 (15.12) | 41.07 (13.54) |
| Estimated number of days taken off because of illness in the last 6 months (SD) | 1.32 (1.88) | 2.46 (4.15) | 1.90 (3.28) |
| Number of children (SD) | 2.08 (1.65) | 2.03 (1.22) | 2.05 (1.44) |
| Years of full-time education (SD) | 15.09 (3.12) | 13.00 (2.81) | 14.03 (3.14) |
| Housing tenure (%) |  |  |  |
| —Mortgage or loan | 46 (71.90) | 49 (74.24) | 95 (73.07) |
| —Rent | 9 (14.10) | 6 (9.10) | 15 (11.53) |
| —Own | 6 (9.40) | 9 (13.60) | 15 (11.53) |
| —Shared ownership | 2 (3.10) | 0 (0) | 2 (1.5) |
| —Live rent free | 0 (0) | 1 (1.57) | 1 (0.76) |
| —Other | 1 (1.60) | 1 (1.57) | 2 (1.5) |
| Marital status (%) |  |  |  |
| —Married and living together | 49 (76.56) | 50 (75.75) | 99 (76.15) |
| —Living together but not married | 4 (6.25) | 6 (9.09) | 10 (7.69) |
| —Single | 3 (4.68) | 5 (7.57) | 8 (6.15) |
| —Separated | 6 (9.37) | 0 (0) | 6 (4.61) |
| —Divorced | 2 (3.05) | 3 (4.54) | 5 (3.84) |
| —Other | 0 (0) | 2 (3.03) | 2 (1.5) |

**Abbreviations:** Aussie-FIT, Australian Fans in Training.

## Outcomes at 3 months: All-cases analysis

The mean difference at 3 months between groups was 3.3 kg (95% CI 1.9–4.8) or 2.88% of total body weight lost (1.48–4.28) from baseline, both in favour of the intervention arm ($p < 0.001$) as per all-cases analysis ($N = 130$); see Table 2. The mean difference in BMI between groups (changes from baseline) was 1.48 kg/m$^2$ (95% CI 0.57–1.73) in favour of the intervention ($p < 0.001$). Mean differences in changes in waist circumference (cm), blood pressure (mm/Hg), step counts, and time spent in sedentary or light activity were not significant between groups. However, the intervention group's MVPA at 3 months was significantly higher than that of the control group by 8.5 min/day (95 CI 1.3, 15.7; $p = 0.02$).

At 3 months, the intervention group reported lower scores for high fat foods (−0.20; 95% CI −0.28, −0.12; $p < 0.001$) and sugar (−0.39; 95% CI −0.66, −0.12; $p = 0.005$) than the control group, but there was no significant improvement in fruit and vegetable score or alcohol intake. In terms of psychosocial outcomes, at 3 months the intervention group, as compared with the control group, reported significantly higher levels of self-esteem (0.23; 95% CI 0.15, 0.29; $p < 0.001$), positive affect (0.46; 95% CI 0.24, 0.69; $p < 0.001$), basic need satisfaction in relation to weight-loss behaviours (0.60; 95% CI, 0.35, 0.84; $p < 0.001$), overall health as per the EuroQol Visual Analogue Scale (14.5; 95% CI 9.7, 19.3; $p < 0.001$), goal facilitation (0.47; 95% CI 0.27, 0.68; $p < 0.001$), habits for physical activity (0.89; 95% CI 0.73, 1.05; $p < 0.001$) and healthy eating (1.08; 95% CI 0.74, 1.42; $p < 0.001$), and planning (0.39; 95% CI 0.20, 0.58; $p < 0.001$). Sleep quality improved from baseline to a greater extent in the intervention than in the control group (−1.18; 95% CI −2.16, −0.20; $p = 0.02$). All other differences on secondary outcomes were small and statistically nonsignificant (Table 2). Intervention (S3 Table) and control (S4 Table) group effects were analysed separately for the 3- and 6-month data.

## Preliminary cost-effectiveness

The cost-effectiveness data were feasible to collect and analyse. The total direct costs associated with the Aussie-FIT program (i.e., program set-up, promotion, and delivery material costs) amounted to AUD\$35,215 (i.e., AUD\$270 per participant), compared with no costs incurred for the comparison group. Thus, the incremental cost per individual was AUD\$270. The preliminary cost-effectiveness of the Aussie-FIT program was assessed using 2 outcome variables: (1) the number of men achieving a 5% weight reduction at 3 months and (2) QALYs. Thus, we calculated a preliminary indication of the incremental cost-effectiveness (ICER) per each additional man achieving a 5% weight loss at 3 months and the incremental cost-effectiveness per QALY.

At 3 months, 19 out of 64 men (approximately 29%) in the intervention group had achieved a 5% weight loss, while only 6 out of 66 men (approximately 9%) in the comparison group had achieved a 5% weight loss. The incremental effect of the program concerning a 5% weight loss at 3 months is estimated to be 0.206 (i.e., 0.297–0.091). Thus, ICER per each additional man achieving a 5% weight loss at 3 months was estimated to be AUD\$1,315 (i.e., $\left(\frac{\$270.888}{0.206}\right)$). The EQ-5D-5L values were converted into utility weights on the basis of the preference weights of a pilot sample from the Australian general population to compute QALYs. At 3 months, the average QALYs for the intervention group were 0.216 ($SD = 0.041$), while for the comparison group, it was 0.209 ($SD = 0.037$). The incremental QALY gains at 3 months were estimated to be 0.007 (i.e., 0.216–0.209). Thus, the cost-effectiveness of the Aussie-FIT was estimated to be AUD\$39,756 (i.e., $\left(\frac{\$270.888}{0.007}\right)$) per an additional QALY gained after 3 months.

**Table 2. Group differences in 3-month changes in primary and secondary outcome (per protocol and all-cases analysis).**

| | Aussie-FIT Intervention Group (Baseline $n = 64$)* | Aussie-FIT Waitlist Control Group (Baseline $n = 66$) | Per Protocol ($n = 112$) | All-Cases Analysis ($N = 130$) |
|---|---|---|---|---|
| | Group mean | Group mean | | |
| Weight (kg)** | 110.15 (16.37) | 112.64 (19.91) | | |
| —3 months | 107.81 (16.58) | 111.21 (20.60) | | |
| —6 months | 107.21 (18.70) | 108.22 (20.93) | −3.92 (−6.10, −1.73), $p < 0.001$ | −3.33 (−4.77, −1.89), $p < 0.001$ |
| Weight (% of total body weight lost since baseline) | | | | |
| —3 months (from T0 to T1) | 3.41 (4.39) | 0.53 (3.19) | | |
| —6 months (from T0 to T2) | 4.51 (6.26) | 3.67 (4.58) | −3.47 (−5.43, −1.51), $p = 0.001$ | 2.88 (−4.28, −1.48), $p < 0.001$ |
| Waist circumference (cm) | 115.23 (11.79) | 117.24 (15.42) | | |
| —3 months | 113.42 (15.87) | 116.08 (15.51) | | |
| —6 months | 108.73 (13.68) | 110.84 (15.46) | −2.08 (−6.60, 1.91), $p = 0.32$ | −2.66 (−5.53, 0.21), $p = 0.07$ |
| BMI (kg/m$^2$) | 34.66 (4.63) | 35.32 (6.53) | | |
| —3 months | 33.81 (5.03) | 35.02 (6.99) | | |
| —6 months | 33.71 (5.46) | 34.33 (6.88) | −1.36 (−2.06, −0.66), $p < 0.001$ | −1.48 (−1.73, −0.57), $p < 0.001$ |
| Blood pressure (mm/Hg) | | | | |
| Systolic | 139.80 (18.81) | 135.95 (14.88) | | |
| —3 months | 136.58 (14.73) | 136.56 (13.13) | | |
| —6 months | 138.29 (14.75) | 136.62 (14.97) | −4.42 (−9.11, 0.26), $p = 0.06$ | −2.16 (−5.72, 1.41), $p = 0.24$ |
| Diastolic | 87.95 (10.31) | 86.67 (9.46) | | |
| —3 months | 87.22 (10.17) | 87.77 (8.64) | | |
| —6 months | 88.69 (7.82) | 88.13 (9.37) | −0.72 (−4.12, 2.67), $p = 0.68$ | −1.09 (−4.51, 2.33), $p = 0.53$ |
| Physical activity measured with ActiGraph | | | | |
| Sedentary time (min/day) | 590.74 (94.55) | 571.9 (101.76) | | |
| —3 months | 650.84 (192.32) | 644.22 (171.81) | | |
| —6 months | 536.32 (96.13) | 561.96 (76.19) | −1.52 (−38.25, 35.20), $p = 0.94$ | −0.95 (−35.18, 33.27), $p = 0.96$ |
| LPA time (min/day) | 261.80 (70.21) | 268.2 (71.7) | | |
| —3 months | 264.72 (67.83) | 276.47 (73.01) | | |
| —6 months | 277.37 (79.24) | 298.59 (75.82) | 2.40 (−29.71, 34.50), $p = 0.88$ | 0.89 (−21.54, 23.31), $p = 0.94$ |
| MVPA time (min/day) | 35.61 (19.57) | 38.38 (21.5) | | |
| —3 months | 45.14 (22.50) | 37.63 (20.45) | | |
| —6 months | 39.90 (22.50) | 51.00 (31.77) | 7.86 (−1.25, 16.97), $p = 0.09$ | 8.54 (1.37, 15.71), $p = 0.02$ |
| Steps (uncensored) | 12,872 (3,770) | 13,097 (3,754) | | |
| —3 months | 13,717 (4,322) | 13,421 (4,047) | | |
| —6 months | 13,762 (4,395) | 15,736 (4,620) | 849 (−972, 2,670), $p = 0.36$ | 445 (−660, 1,551), $p = 0.43$ |
| Steps (censored) | 10,787 (3,359) | 11,053 (3,557) | | |
| —3 months | 11,654 (4,018) | 11,356 (3,828) | | |
| —6 months | 11,718 (4,049) | 13,439 (4,407) | 777 (−561, 2,115), $p = 0.26$ | 550 (−552, 1,652), $p = 0.33$ |
| Self-reported outcome measures | | | | |

(*Continued*)

**Table 2.** (*Continued*)

| | Aussie-FIT Intervention Group (Baseline *n* = 64)* | Aussie-FIT Waitlist Control Group (Baseline *n* = 66) | Per Protocol (*n* = 112) | All-Cases Analysis (*N* = 130) |
|---|---|---|---|---|
| | Group mean | Group mean | | |
| Fatty food score (DINE) | 1.91 (0.34) | 1.91 (0.31) | | |
| —3 months | 1.70 (0.29) | 1.89 (0.33) | | |
| —6 months | 1.74 (0.29) | 1.61 (0.29) | −0.19 (−0.32, −0.06), $p < 0.001$ | −0.20 (−0.28, −0.12), $p < 0.001$ |
| Fruit and vegetable (DINE) | 1.87 (0.37) | 1.85 (0.53) | | |
| —3 months | 2.24 (0.56) | 2.03 (0.56) | | |
| —6 months | 2.20 (0.47) | 2.28 (0.64) | 0.18 (−0.06, 0.42), $p = 0.15$ | 0.20 (0.01, 0.39), $p = 0.036$ |
| Sugary food score (DINE) | 1.39 (0.36) | 1.44 (0.47) | | |
| —3 months | 1.10 (0.16) | 1.51 (0.56) | | |
| —6 months | 1.16 (0.29) | 1.16 (0.39) | −0.39 (−0.61, −0.18), $p < 0.001$ | −0.39 (−0.66, −0.12), $p = 0.005$ |
| Total alcohol consumption (units per week) | 8.70 (11.34) | 8.44 (10.88) | | |
| —3 months | 11.50 (8.54) | 8.54 (6.09) | | |
| —6 months | 7.04 (8.95) | 5.81 (5.43) | 0.42 (−0.82, 1.66), $p = 0.50$ | 1.06 (−0.99, 3.11), $p = 0.31$ |
| Self-esteem (Rosenberg scale, range 1–4) | 3.02 (0.51) | 2.98 (0.37) | | |
| —3 months | 3.23 (0.51) | 3.03 (0.44) | | |
| —6 months | 3.21 (0.54) | 3.25 (0.42) | 0.13 (0.02, 0.24), $p = 0.02$ | 0.15 (0.05, 0.25), $p = 0.003$ |
| Positive affect (PANAS, range 1–5) | 2.92 (0.70) | 2.89 (0.66) | | |
| —3 months | 3.56 (0.64) | 2.97 (0.66) | | |
| —6 months | 3.35 (0.58) | 3.59 (0.63) | 0.58 (0.37, 0.79), $p < 0.001$ | 0.58 (0.35, 0.80), $p < 0.001$ |
| Negative affect (PANAS, range 1–5) | 1.71 (0.60) | 1.75 (0.56) | | |
| —3 months | 1.53 (0.51) | 1.69 (0.58) | | |
| —6 months | 1.59 (0.54) | 1.48 (0.59) | −0.11 (−0.29, 0.07), $p = 0.25$ | −0.14 (−0.30, 0.02), $p = 0.09$ |
| Need support (IBQ scale, range 1–7) | 5.07 (1.18) | 5.33 (1.07) | | |
| —3 months | 5.35 (1.27) | 5.43 (1.13) | | |
| —6 months | 5.33 (1.09) | 5.64 (1.21) | 0.18 (−0.69, 1.06), $p = 0.68$ | 0.15 (−0.43, 0.73), $p = 0.61$ |
| Basic need satisfaction in relation to weight-loss behaviours (range 1–5) | 3.23 (0.59) | 3.40 (0.65) | | |
| —3 months | 3.92 (0.61) | 3.40 (0.72) | | |
| —6 months | 3.73 (0.75) | 3.97 (0.66) | 0.72 (0.50, 0.94), $p < 0.001$ | 0.60 (0.35, 0.84), $p < 0.001$ |
| Weight-loss motivation (range 1–5) | | | | |
| Autonomous | 4.32 (0.53) | 4.22 (0.55) | | |
| —3 months | 6.32 (0.49) | 6.23 (0.55) | | |
| —6 months | 6.33 (0.67) | 6.50 (0.45) | 0.14 (−0.05, 0.32), $p = 0.14$ | 0.07 (−0.12, 0.24), $p = 0.48$ |
| Controlled | 2.79 (0.65) | 2.91 (0.58) | | |
| —3 months | 4.26 (1.22) | 4.28 (0.97) | | |
| —6 months | 3.97 (1.07) | 4.10 (1.21) | 0.04 (−0.46, 0.54), $p = 0.88$ | 0.11 (−1.22, 1.44), $p = 0.87$ |
| Health-related quality of life (health utility score) | 0.86 (0.20) | 0.82 (0.16) | | |

(*Continued*)

**Table 2.** (Continued)

| | Aussie-FIT Intervention Group (Baseline $n = 64$)* | Aussie-FIT Waitlist Control Group (Baseline $n = 66$) | Per Protocol ($n = 112$) | All-Cases Analysis ($N = 130$) |
|---|---|---|---|---|
| | Group mean | Group mean | | |
| —3 months | 0.89 (0.13) | 0.84 (0.16) | | |
| —6 months | 0.86 (0.16) | 0.86 (0.19) | 0.02 (−0.02, 0.07), $p = 0.36$ | |
| Total for health today (range 0–100) | 59.28 (18.63) | 54.92 (16.65) | | |
| —3 months | 74.15 (15.10) | 57.66 (15.82) | | |
| —6 months | 75.37 (13.28) | 69.50 (14.62) | 16.10 (13.22, 18.97), $p < 0.001$ | 14.57 (9.77, 19.34), $p < 0.001$ |
| Goal facilitation (range 1–5) | 2.64 (0.76) | 2.80 (0.75) | | |
| —3 months | 3.17(0.91) | 2.81 (0.85) | | |
| —6 months | 3.19 (0.75) | 3.41 0.79) | 0.54 (−0.09, 1.16), $p = 0.09$ | 0.47 (0.27, 0.68), $p < 0.001$ |
| Competing goals (range 1–5) | 3.80 (0.65) | 3.67 (0.72) | | |
| —3 months | 3.38 (0.68) | 3.52 (0.71) | | |
| —6 months | 3.39 (0.75) | 3.04 (0.92) | −0.20 (−0.48, 0.07), $p = 0.15$ | −0.16 (−0.45, 0.13), $p = 0.27$ |
| Habits for PA (range 1–7) | 3.18 (1.14) | 3.44 (1.27) | | |
| —3 months | 3.96 (1.31) | 3.30 (1.30) | | |
| —6 months | 4.01 (1.30) | 4.18 (1.40) | 1.07 (0.70, 1.45), $p < 0.001$ | 0.89 (0.73, 1.05), $p < 0.001$ |
| Habits for eating (range 1–7) | 3.02 (1.22) | 3.01 (1.07) | | |
| —3 months | 4.01 (1.30) | 2.99 (1.24) | | |
| —6 months | 3.78 1.35) | 4.05 (1.21) | 1.17 (0.66, 1.57), $p < 0.001$ | 1.08 (0.74, 1.42), $p < 0.001$ |
| Barriers (range 1–5) | 3.16 (0.90) | 3.28 (0.69) | | |
| —3 months | 3.16 (0.78) | 3.14 (0.70) | | |
| —6 months | 3.09 (0.81) | 3.36 (0.83) | 0.21 (−0.07, 0.48), $p = 0.14$ | 0.07 (−0.08, 0.21), $p = 0.38$ |
| Planning (range 1–5) | 3.00 (0.83) | 3.37 (0.86) | | |
| —3 months | 3.70 (0.68) | 3.39 (0.71) | | |
| —6 months | 3.40 (0.92) | 3.82 (0.82) | 0.52 (0.26, 0.78), $p < 0.001$ | 0.39 (0.20, 0.58), $p < 0.001$ |
| Sleep quality (PSQI, range 0–21; lower score indicates a better outcome) | 5.76 (3.05) | 6.59 (3.59) | | |
| —3 months | 4.11 (2.30) | 5.66 (3.02) | −1.29 (−2.03, −0.55), $p = 0.001$ | −1.18 (−2.16, −0.20), $p = 0.02$ |
| —6 months | 4.77 (3.30) | 4.14 (3.03) | | |

*$n = 50$ for 3-month measure for intervention group and $n = 62$ for 3-month measure control group; $n = 35$ for 6-month measure for intervention group and $n = 47$ for control group at T2.

**Top line is a baseline measure. Group means are completer's means. Censored step counts (n/day) are any steps taken during minutes in which the ActiGraph cpm were >100; variance for step count data was larger than the maximum permitted in Mplus (1,000,000), so the dependent variable and dependent variable at baseline (which we included in the model as a covariate) were rescaled by 10 (i.e., linear transformation) for censored and uncensored steps. **Abbreviations:** Aussie-FIT, Australian Fans in Training; BMI, Body Mass Index; cpm, counts per minute; DINE, dietary instrument for nutritional education; IBQ, Interpersonal Behaviours Questionnaire; LPA, Leisure Time Physical Activity; MVPA, moderate-to-vigorous physical activity; PANAS, Positive and Negative Affect Scale; PSQI, Pittsburg Sleep Quality Index.

## Discussion

### Summary of findings

This pilot RCT aimed to determine the feasibility of delivering and evaluating the 12-week Aussie-FIT program in the context of AFL in Australia to determine the appropriateness of a future RCT. Findings support the feasibility of delivering this intervention in this context and its preliminary efficacy. Men who participated in the Aussie-FIT program lost a significant amount of weight and sustained this weight loss 3 months postintervention. Pilot data also indicated that the program had positive effects on a range of behavioural and psychosocial outcomes. The recruitment process was very effective, with many more men applying than we had capacity to offer the program to as part of this study.

Similar to the experiences of FFIT in Scotland, capitalising on the high popularity of the 2 participating AFL clubs in Perth proved to be an effective way to attract men to the program. Trial retention at the 3-month follow-up was excellent and comparable with other FFIT trials. However, there was quite high loss at 6 months, with only 63% of the original sample retained. Overall, these findings support the appropriateness of a future fully powered RCT to more rigorously test the effectiveness and cost-effectiveness of the Aussie-FIT intervention as a means to address overweight and obesity among men in Australia.

### Relationship to other literature

Our findings for weight loss at 3 months are comparable to most other gender-targeted interventions delivered in sports settings internationally. The results of our pilot study are comparable to those from the HockeyFIT pilot trial in Canada [18] ($N = 80$) and the RuFIT pilot in New Zealand [17] ($N = 96$), which reported mean differences in weight loss between groups at 3 months to be 3.6 kg (95% CI 1.9–5.3) and 2.5 kg (95% CI −0.4 to 5.4), respectively, in favour of the intervention. Compared to the FFIT ($N = 748$) trial, Aussie-FIT participants' weight loss immediately post-program was lower; FFIT participants lost 5.2 kgs (−6 to −4.3) at 3 months [14]. However, any comparisons with FFIT must be interpreted with caution, given that the FFIT evaluation was not a pilot.

In relation to physical activity outcomes, we found only a small increase in step counts following participation in the program for the intervention arm; however, this warrants more rigorous testing via a fully powered RCT, given the relatively large proportion of participants with step counts meeting general PA guidelines at baseline. The FFIT study used a self-report measure of physical activity, whereas the recent EuroFIT intervention reported device-measured physical activity (steps) as the main outcome; intention to treat analyses in EuroFIT showed a baseline-adjusted mean difference of 1,208 steps per day (95% CI, 869–1,546) in favour of the intervention group at 3 months. It is difficult to explain the high baseline in step count in this study. Anecdotally, we are aware that many participants were in manual professions (for example, construction work); hence, it is possible that there was a sampling bias, However, we did not collect data on occupations, so we cannot draw conclusions as to a potential bias. We also note that the ActiGraph algorithm classifies intermittent stepping more readily than the ActivPAL algorithm (used in the EuroFIT trial),which is shown to result in a nontrivial discrepancy between devices [64].

In line with other FIT studies [14,20], Aussie-FIT participants successfully decreased their consumption of fatty and sugary foods following the program, and point estimates indicated small increases in fruit and vegetable intake; however, it is noteworthy that scores were already relatively high at the start of the intervention. Excessive alcohol consumption is often perceived to be central to the culture of football spectatorship in Australia, and the program

specifically aimed to reduce it [65]. In the current sample, men reported low alcohol intake on average at baseline, which was below the alcohol guideline limits for Australian males. It is possible that men underreported alcohol consumption or that this study attracted a sample of men who were moderate drinkers.

The inclusion of motivational components in the coach training and program content of Aussie-FIT were a key adaptation and extension of previous trials. There is preliminary evidence to suggest that participating in the Aussie-FIT program led participants to experience increased feelings of autonomy, competence, and relatedness in relation to weight-loss behaviours. These findings are important because, according to SDT [25], feeling that one is competent, autonomous, and related to others are critical determinants of adaptive behaviours, cognitions, and emotions [66]. In a future RCT, longer-term follow-ups will help to determine whether participation in Aussie-FIT leads to longer-term experience of need satisfaction and/or whether experiences of need support offered during Aussie-FIT are associated with better outcomes for participants.

We also assessed constructs that were targeted in the intervention as relevant for weight loss but were not previously addressed in FIT-based studies. For instance, there was a significant improvement in goal facilitation. This tentatively suggests that Aussie-FIT participants learnt how to prioritise their weight loss. However, there were no important/significant effects for dealing with competing goals. This may be due to a low emphasis in the program content on dealing with other competing goals [67]. The improvements in self-reported behavioural automaticity for physical activity and healthy eating were consistent with findings from other studies promoting habit formation [68]. There was also improvement in planning at 3 months in favour of the intervention group, consistent with previous research on forming action and coping plans [56,57].

Finally, there was improved self-reported sleep quality at 3 months in favour of the intervention group. Given the burgeoning concerns with the ill-health effects of bad quality of sleep [69], this result is promising. This finding is based on well-validated self-report instrument but warrants replication with device-measured monitoring of sleep quality. Other FIT studies did not examine sleep change as an outcome of the intervention. Sleep duration and quality have an important influence on physical activity levels and consequently on weight. Better sleep is correlated with higher physical activity and lower weight [70].

Our findings support progression to a fully powered RCT to further test the effectiveness of the Aussie-FIT program; a future trial should be powered with data from this pilot rather than the FFIT study and also be powered to detect changes in secondary outcomes. The long-term goal of the Aussie-FIT program is to promote long-term maintenance of behaviour changes that have led to weight loss; therefore, longer-term follow-up studies are required to determine whether specific additions included in this program have an impact on long-term outcomes. For instance, FFIT program participants maintained intervention effects at 3.5 years postbaseline (2.9 kg [95% CI 1.7, 4 kg], $p < 0.001$) [15]; further exploration is needed to assess if the inclusion of specific SDT techniques and behaviour change techniques adds value that corresponds with further weight reduction or weight maintenance at a desired level.

We intended to recruit coaches via the AFL clubs; however, the clubs did not have coaches readily available to fulfil this role. As a result, 3 coaches were recruited via recommendations from the clubs, and 3 coaches were independently identified by the research team. All Aussie-FIT coaches' professional backgrounds (which included teaching, coaching, exercise instruction, and sport science) meant they were likely to be equipped to create an environment that is seen as central to the success of FFIT, i.e., a nondidactic, encouraging, and interactive delivery style that incorporates appropriate banter to support vicarious learning and team spirit [71]. However, those coaches who were not directly associated with the club may have been less able

to integrate into their coaching style other characteristics seen as contributing to FFIT's success, such as the 'behind-the-scenes' stories and tacit knowledge of the inner workings of the club into program delivery [71]. To overcome this, within the coach training, the coaches were encouraged to think about how to incorporate the sport and club 'feel' within their delivery style. Based on qualitative data, the level of connection between the coach and the club did not seem to contribute to any variability in quality of experience any more than other relevant variables such as the coaches' personality and motivation or differences in facilities available to deliver the program. However, the extent to which the personal and situational characteristics that shape program delivery impact participants' experiences of the program and observed outcomes could be further explored in future research to help inform the most appropriate implementation model for Australia.

The preliminary cost-effectiveness evaluation suggests that the Aussie-FIT program was relatively inexpensive to deliver and potentially cost-effective, with estimated ICERs of $1,315 per each additional man achieving a 5% weight loss and $39,756 per QALY gained after 3 months. Moreover, our calculated ICER per QALY lies within the commonly acceptable range considered to be good value for money in Australia [72,73]. However, the reported ICER in this pilot study should be interpreted with caution and may be overestimated because it does not include several relevant costs, such as direct medical costs and future health system costs due to the intervention. A complete and comprehensive cost-effectiveness evaluation using the model piloted in this study is warranted in a fully powered trial.

## Strengths, limitations, and future directions

This study had several strengths. It was the first trial to assess an FIT intervention in Australia in the AFL setting. Key strengths also include the evidence of effectiveness of strategies to recruit men, feasible trial procedures such as the use of new questionnaires and device-based assessments, and feasible incorporation of principles of effective motivation into the intervention content. The limitations of the study include the inability to collect attendance data at the Aussie-FIT weekly sessions and low retention at final follow-up. One possible explanation for this loss could be the timing of data collections. Because of unavoidable time restrictions on when the program could be delivered, final follow-ups were scheduled the week before Christmas, which is usually a busy time of personal commitments and festivities and may have prevented men from being able to attend. Unfortunately, we were unable to collect reliable data on men's attendance at the program as coaches only sporadically recorded attendance. This was likely because coaches were busy at the start of the program, talking to men and preparing for the session. In a future trial, it may be more advantageous to use a 'self-check–in' system on an iPad, on which men can register when arriving at the venue. This study also lacked objective measurement of sleep quality, body composition, and cardiorespiratory or neuromuscular fitness. A fully powered future trial could include these measurements to further improve quality of the outcome assessment. Also, some of the measured outcomes did not improve significantly in the intervention group at 3 months (for example, waist circumference, blood pressure, sedentary time, need support). Nevertheless, findings were consistently in favour of the intervention group. To fully understand whether there are no important/significant differences, a fully powered trial needs to be conducted.

Another limitation of the current study is the generalisability of findings with respect to ethnicity because the sample was not representative of the Australian population (95% were White). Future formative work should include feedback on resources from a diverse sample of men. Future trials should investigate whether findings can be replicated in other ethnic groups or whether the program requires additional tailoring to appeal to a more diverse sample of

men in Australia (for example, men from culturally and linguistically diverse backgrounds, indigenous men, men from across the socioeconomic spectrum). Because there are only a limited number of AFL clubs in Australia, a fully powered RCT may need to also rely on lower-level clubs (for example, state-based leagues) to reach target numbers and attract men from a wider geographical reach.

## Conclusion

The pilot findings may be generalizable to delivery of the intervention in other professional sport settings in Australia, and broader implementation may be better achieved by capitalising on rollouts of the program in lower-level state-based leagues and via adaptations to other sports. The Aussie-FIT model also has potential for expansion via delivery to specific population segments who are not necessarily overweight or obese but are insufficiently active to benefit health and for whom existing interventions may not appeal (for example, new dads, prostate cancer survivors). Pilot studies are needed to determine whether tailoring the program to engage these specific population groups can offer an alternative or superior opportunity to 'usual care'. Overall, this study achieved the objectives of testing the feasibility of delivering and evaluating Aussie-FIT in the AFL context in Australia. Recommendations to use these methods in a future definitive RCT are warranted. The Aussie-FIT program should be tested and implemented on a larger scale with maintenance effects tested over a longer timeframe (for example, 1- and 2-year follow-ups).

## Supporting information

**S1 Table. CONSORT guidelines extension for randomised pilot and feasibility trials and TIDieR checklists.** (S1A) CONSORT 2010 checklist of information to include when reporting a pilot or feasibility trial. (S1B) The TIDieR Checklist. CONSORT, Consolidated Standards of Reporting Trials; TIDieR, Template for Intervention Description and Replication.
(DOCX)

**S2 Table. Accelerometer data at all time points for participants with 4+ days of valid (i.e., ≥10 h/day) wear time; data are means (SD).**
(DOCX)

**S3 Table. Changes from 3 months to 6 months for the intervention group only.**
(DOCX)

**S4 Table. Program effects (i.e., change from 3 months to 6 months) for the waitlist control group.**
(DOCX)

**S1 Text. All scales reliability checks.** (SPSS file, here: https://osf.io/4vsng/files/).
(DOCX)

## Acknowledgments

Aussie-FIT builds on the FFIT program, the development and evaluation of which was undertaken by a research team led by the University of Glasgow with funding from various grants including a Medical Research Council (MRC) grant (reference number MC_UU_12017/3), a Chief Scientist Office (CSO) grant (reference number CZG/2/504), and a National Institute for Health Research grant (NIHR) (reference number 09/3010/06). The development and evaluation of FFIT was facilitated through partnership working with the Scottish Professional

Football League Trust (SPFLT). We would like to thank the AFL fans ($N$ = 130) who participated in the Aussie-FIT study, our 6 coaches, the participating AFL clubs, and the team of students and research assistants from Curtin University and Edith Cowan University.

## Author Contributions

**Conceptualization:** Dominika Kwasnicka, Nikos Ntoumanis, Kate Hunt, Cindy M. Gray, Robert U. Newton, Daniel F. Gucciardi, Cecilie Thøgersen-Ntoumani, Deborah A. Kerr, Sally Wyke, Philip J. Morgan, Suzanne Robinson, Eleanor Quested.

**Data curation:** Dominika Kwasnicka, Eleanor Quested.

**Formal analysis:** Dominika Kwasnicka, Nikos Ntoumanis, Daniel F. Gucciardi, Joanne McVeigh, Suzanne Robinson, Marshall Makate, Eleanor Quested.

**Funding acquisition:** Nikos Ntoumanis, Kate Hunt, Cindy M. Gray, Daniel F. Gucciardi, Cecilie Thøgersen-Ntoumani, Deborah A. Kerr, Sally Wyke, Philip J. Morgan, Suzanne Robinson, Eleanor Quested.

**Investigation:** Dominika Kwasnicka, Nikos Ntoumanis, Robert U. Newton, Daniel F. Gucciardi, Cecilie Thøgersen-Ntoumani, Jenny L. Olson, Joanne McVeigh, Deborah A. Kerr, Eleanor Quested.

**Methodology:** Dominika Kwasnicka, Nikos Ntoumanis, Kate Hunt, Cindy M. Gray, Robert U. Newton, Daniel F. Gucciardi, Cecilie Thøgersen-Ntoumani, Joanne McVeigh, Deborah A. Kerr, Sally Wyke, Philip J. Morgan, Suzanne Robinson, Eleanor Quested.

**Project administration:** Dominika Kwasnicka, Jenny L. Olson, Eleanor Quested.

**Resources:** Dominika Kwasnicka, Nikos Ntoumanis, Robert U. Newton, Eleanor Quested.

**Software:** Joanne McVeigh.

**Supervision:** Dominika Kwasnicka, Eleanor Quested.

**Writing – original draft:** Dominika Kwasnicka, Eleanor Quested.

**Writing – review & editing:** Dominika Kwasnicka, Nikos Ntoumanis, Kate Hunt, Cindy M. Gray, Robert U. Newton, Daniel F. Gucciardi, Cecilie Thøgersen-Ntoumani, Jenny L. Olson, Joanne McVeigh, Deborah A. Kerr, Sally Wyke, Philip J. Morgan, Suzanne Robinson, Marshall Makate, Eleanor Quested.

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
