## [Decision Letter · Decision Letter 0]

24 Feb 2020

Dear Dr. Quested,

Thank you very much for submitting your manuscript "A gender-sensitised weight loss and healthy living program for overweight and obese men in Australian Football League settings (Aussie-FIT): A pilot randomised controlled trial." (PMEDICINE-D-19-03524) for consideration at PLOS Medicine. I sincerely apologise for the delay in getting this decision to you. 

[LINK]

In light of these reviews, I am afraid that we will not be able to accept the manuscript for publication in the journal in its current form, but we would like to consider a revised version that addresses the reviewers' and editors' comments. Obviously we cannot make any decision about publication until we have seen the revised manuscript and your response, and we plan to seek re-review by one or more of the reviewers. 

We expect to receive your revised manuscript by Mar 09 2020 11:59PM. Please email us (plosmedicine@plos.org) if you have any questions or concerns.

We look forward to receiving your revised manuscript. 

Sincerely,

Adya Misra, PhD

Senior Editor 

PLOS Medicine

plosmedicine.org

Abstract- please use an alternative to “hook” 

Abstract- please use specific language instead of “meaningful differences”. If you choose to report your secondary outcomes here, we suggest you include 95% CI and p values. 

Abstract methods and findings- the last sentence of this section should highlight a limitation of your study design

Please include the study protocol document and analysis plan, with any amendments, as Supporting Information to be published with the manuscript if accepted. If this is published, please provide clear details in the methods section. 

References must be within square brackets throughout 

Please provide the primary and secondary outcome measures within the methods section. A brief summary of the trial outcomes can be presented in the introduction section 

Questionnaires -please provide the questionnaires used in this study as SI files or if previously published, please provide a citation

Please provide 95% confidence intervals along with all p-values, all p-values should be in 0.xx format

Discussion- I recommend that the results outlined at line 448 are reported in the results section 

The primary outcomes on your trial registry do not fully match the primary outcomes reported in the manuscript, For instance- acceptability to participants and coaches. If a clarification is needed in the submission, please revise. 

Please include sections and paragraph numbers instead of page numbers in the CONSORT checklist

The Data Availability Statement (DAS) requires revision. For each data source used in your study: 

Comments from the reviewers:

Reviewer #1: This manuscript reports the results from a pilot of the Aussie-FIT program. The manuscript is well written and descriptive and I only have a couple of minor comments. 

Line 105: Please change to "...healthy habits to maintain long-term behaviour change."

Lines 115-124: You do not need to list all the outcomes here - just a brief summary is sufficient for the introduction.

Lines 173-175: Was feedback on the program only from 5 men? This is very small, and I would consider adding to the discussion that more formative work may have helped make the program more culturally appropriate for Australian men.

Line 181: Please state who delivered the training workshops.

Lines 349-350: Please add the proportions of men (i.e. in %) for follow-up measures.

Lines 372-373: When reporting the results, please provide more information as it is unclear exactly what you are measuring here i.e. what does the 2.88% refer to? I see this is more clear in the table but still needs to be written out in full in the text.

Line 390: Are you referring to sleep quality here, because as it stands I initially interpreted it as sleep duration (in which case decreasing sleep duration would not be a positive outcome!) Given that the sentence opens with "...reported significantly higher levels of..." this still refers to the sleep outcome (which decreased). This is very confusing. To aid understanding, I suggest including the sleep quality measure as its own sentence i.e. Sleep quality improved from baseline to a greater extent in the intervention versus control group. And I would update the table reporting sleep outcomes to indicate the a lower score indicates a better outcome.

Discussion: I think you need to discuss the generalisability of findings with respect to ethnicity - given that over 95% of participants were Caucasian, the sample isn't representative of the Australian population - and future work would need to investigate whether findings can be replicated in other groups - formative work may be needed to establish how the program could be tailored appropriately. 

Reviewer #2: Colin Greaves review: Accept with Minor Revision

Overall, this is a very clearly written and high quality report of an important pilot study around the adaptation /piloting of the highly successful FFIT intervention for implementation in an Aussie Rules context in Western Australia. I have only some minor, mainly presentational comments (and a more substantive comment about health economics, which is made based on exposure to health economic evaluations in several full-scale trials, although I am not an expert in this area)

Abstract: Could mention 3-6 mth results (see below comments on presentation of these). One decimal place might be sufficiently precise?

BACKGROUND:

"Men need access to plentiful resources to maintain their behaviour long term, including both physical (access to healthy food, exercise 108 equipment) and psychological 109 (energy levels, positive mood) resources to facilitate health behaviour change maintenance 110 (28,29)." ... and social resources?

METHODS

Sample size: 

"The study sample size followed guidelines for pilot RCTs (30,31) and approximated the FFIT pilot study sample size (32). The Aussie-FIT pilot study was designed to inform the development of a definitive future trial.

- Given the additional /non-standard pilot study aim to 'assess preliminary efficacy', you should present a formal sample size calculation (based on detecting a difference in weight loss). It is not sufficient to simply refer to other pilot studies as a justification for the sample size selected. Even if the aims were only to estimate retention rates, a sample size calculation can be presented (e.g. calculate N needed to estimate retention rates with CIs of +/- 8%)

Maybe say something about the background experience /qualifications of the facilitators? (in Discussion, how does this compare with FFIT and how might this have affected performance /interactions with participants?)

Intervention: what content /contact (if any) beyond 12 weeks? important to know given the emphasis on maintenance here.

"We measured health-related quality of life using the EuroQol five-dimensional five level version (EQ-5D-5L, range 1-5), with lower scores indicative of higher health-related quality of life."

- How was a single /overall QoL score computed?

Analysis: 

How were missing data imputed for the ITT analysis? e.g. baseline carried forward, LOCF?

I am a bit confused by the logic behind the secondary analysis which seems to combine "maintenance effects for the intervention group and treatment effects for the control group". So, by combining, it measures neither of these things? It might be better to present the 3-6 month within-group changes separately? Not as a comparison - just mean differences with 95%CIs?

I also don't follow the following "We modelled the linear effect of time (3 to 6 months) as a predictor of dependent variables, adjusted for baseline values and the clustering effect of group. In so doing, the effect can be interpreted as the average amount of change on the dependent variable between 3 and 6 months." I'm not a statistician, but I cant see how that interpretation arises. It might be better just to lose this analysis (as above)?

RESULTS:

Baseline steps and MVPA are high? It is a bit hard to say anything definitive about MVPA as unbouted MVPA is usually 5-10 times higher than MVPA in ten-minute bouts, depending on the sample. Steps do look high though, suggesting there may be some sample bias here - so it may be worth picking this up in the limitations - and say how you might seek to ameliorate this in the full-scale trial?

3 month outcomes: Pease make clear that the analyses presented in the first and second paragraphs (3.33 kg (95% CI 1.89-4.77) etc) are the ITT (all participants with imputed values) figures. You could also summarise the per protocol data here?

Is "overall health the EQ-5D-5L score (and if so, which one - the thermometer or other summary score)?

3-6 month outcomes: As above, the rationale for this analysis does not make sense - it combines initial weight loss in the control group with post-intervention maintenance in the intervention group and uses a rather odd (to me) statistical approach (estimating 3-6 month change for the combined groups, whilst controlling for baseline values). Why not simply report the 3-6 month changes within each group (mean change = ***kg, 95%CI: *** to ***). The best estimate of 3-6 maintenance would be the within intervention group change from 3 to 6 months, where weight appears to further decrease by around 0.6kg (Table 2). Also need to report this figure with and without imputation of missing values.

Cost-effectiveness: The calculations here seem to be more back-of-the-envelope than based on professional health economic analytic methods? I am not a health economist, but I do know that a) you have to include health-care usage and other societal costs (depending on the perspective taken) in the "cost" side of the equation (for both groups) and b) you don't just subtract utility values and divide the cost by this difference - there is a statistical analysis of individual level data involved which yields 95%CIs around the estimate. NB: I wouldn't expect to see a good ICER for within-trial analysis of EQ-5D data in a weight loss trial. The sensitivity of EQ-5D is too low and the main QoL benefits (not getting diabetes, avoiding CV events, cancer etc) that will actually influence EQ-5D tend to accrue in the longer term. A decision-analytic modelling approach would therefore be more appropriate for the main trial. 

Furthermore, it doesn't make sense to try to construct a full-blown health economic analysis based on only 110 participants and a 3-month timeframe. I would remove this health economic analysis and just report intervention costs, any between group difference in EQ-5D at 3 months (as part of Table 2) and the measures-completion rate - which indicates whether or not a health ec. analysis would be feasible for the main trial.

Similarly, I would remove the probabilistic sensitivity analysis - the data is much too thin (low N) to bear this type of analysis?

DISCUSSION

This is quiet long /rambling. It could benefit form some sub-headings to structure the text: Summary of findings; Relationship to other literature, Strengths and Limitations; Future research directions; Conclusion

I would cut /substantially edit the health economics text (as per comments above)

TABLES

In general, please consider the appropriate number of significant figures /decimal places. Are your estimates really so accurate as to justify reporting of age, weight etc to 2 decimal places?

Table 2: Are the means reported at 3 and 6 months imputed means (for all 64/66 participants) or per protocol means (for responders only) - please indicate

Overall, this is a very worthy paper and a pleasure to read. Good luck with the revisions

Reviewer #3: Alex McConnachie, Statistical Review

Kwasnicka and colleagues present a report of a pilot RCT of a weight loss and healthy lifestyle intervention delivered in Australian Football League clubs. This review considers the statistical aspects of the paper.

I note that the analysis is described as intention to treat, including everyone who provided baseline data. It is stated that ITT analysis "was used to deal with missing data". In my opinion, ITT analysis has nothing to do with missing data. As the name suggests, I would say that ITT is about which intervention it was intended to deliver. In other words, ITT means to analyse according to randomised group, regardless of whether the intervention was received or not. Missing data is a separate issue. Given the description of the statistical methods, I assume that the analysis involved analysing data from all time points, including baseline, in a mixed effects regression model. This accounts for missing follow-up data to some extent, and is a perfectly reasonable way to carry out the analysis. My only concern is the description of this being "an ITT analysis"; I would separate the methods used to deal with missing data from the concept of ITT.

The 5 items of the EQ-5D questionnaire appear to have been averaged to give a score between 1 and 5. This is not usual; the responses to the 5 questions should be mapped to a health utility score at each time point. It is these health utilities that are used to derive QALYs.

Some of the data has been reported over-precisely. E.g. from line 355 onwards, we have summaries of a number of measures (weight, BP, MVPA) reported to 2 decimal places.

Perhaps a little too much emphasis is put onto the statistical significance (or lack of) for between-group comparisons of some of the outcomes. More emphasis should be put on the point estimates of intervention effects (and their CIs). As the authors note at one point, in a pilot trial, the power will be limited to detect important differences between groups. It appears that the majority of measures have point estimates for the between-group difference at 3 months that are in favour of the intervention group. Some of the estimates (e.g. waist circumference, estimated mean difference -2.66cm) are quite sizeable, so whilst they are not statistically significant, I would say that these estimates add to the overall picture of clinically important intervention effects.

Similar comments apply to the changes between 3 and 6 months - a point estimate of an additional 0.98kg weight loss after the end of the intervention is more than just a lack of weight regain - to me, this suggests that the 3-month intervention may continue to have beneficial effects after the end of intervention delivery. This is what would be hoped of a lifestyle intervention, and though the change was not statistically significant in this pilot trial, it is promising, and lends support to the plan to do a larger trial.

This may be a matter for the journal editors, but I prefer to read "0.xx", rather than ".xx".

[LINK]

---

## [Decision Letter · Decision Letter 1]

16 Apr 2020

Dear Dr. Quested,

Thank you very much for re-submitting your manuscript "A gender-sensitised weight loss and healthy living program for overweight and obese men in Australian Football League settings (Aussie-FIT): A pilot randomised controlled trial." (PMEDICINE-D-19-03524R1) for review by PLOS Medicine.

I have discussed the paper with my colleagues and the academic editor and it was also seen again by xxx reviewers. I am pleased to say that provided the remaining editorial and production issues are dealt with we are planning to accept the paper for publication in the journal.

[LINK]

We look forward to receiving the revised manuscript by Apr 23 2020 11:59PM. 

Sincerely,

Adya Misra, PhD

Senior Editor 

PLOS Medicine

plosmedicine.org

Requests from Editors:

Abstract

Last section of the methods and findings section must highlight a limitation of your study methodology

Please include dates of recruitment and intervention in the abstract

Data availability statement- please remove “option C applies”.

CONSORT checklist-please remove page numbers and use sections and paragraphs. The page numbers are likely to change during the publication process.

Some of the references towards the end of the introduction are in rounded brackets. Please change to square brackets

Please use Vancouver style for references 

Please address the comments from the statistical reviewer regarding power analysis and EQ-5D. This involves removing the post-hoc power analysis and also remove the summary score for EQ-5D. We will be unable to accept your manuscript without this. 

Please change all p-values to 0.xx instead of .xx

We would be grateful if you could provide a clean version of your manuscript and one containing track changes as a companion file. 

Suggest rephrasing “meaningful” in the abstract and the discussion as it is too vague for a research article

Line 581 in discussion, please rephrase “footy fandom”

Some of the information in the Acknowledgements regarding grants should be provided in the funding disclosure

Comments from Reviewers:

Reviewer #2: I am happy with the responses provided. No further comments.

Reviewer #3: Alex McConnachie, Statistical Review

I would like to thank the authors for considering my earlier points. However, I feel that the changes they have made have not all been for the better.

In response to another reviewer, a sample size calculation has been added. Previously, as in the protocol paper, the sample size was said to be based on guidelines for pilot trials. This was fine, and was appropriately referenced. The new, post-hoc sample size calculation is not required, and is incorrect. To have 90% power, at 5% significance, to detect a difference of 4.71, assuming a SD of 3.95, requires 2 x 10.51 x (3.95)^2 / (4.71)^2 = 15 per group. If 12 per cluster are followed up (80% of 15), and the ICC is 0.05, the design effect is 1 + 0.05 x (12-1) = 1.55, increasing the sample size to 24 per group. Dividing by 0.8, to allow for loss to follow-up, gives a total of 30 per group randomised.

In my opinion, the sample size justification should be reported in the way the study was designed. If this was based on published guidelines for studies of this type, so be it. A post-hoc power calculation is not required, and is generally not recommended.

One area where I agree with the other reviewer is the number of decimal places used to summarise many of the results. The authors state that they have followed the journal style, though I could find no reference to the number of decimal places to use on the journal website, and "APA guidelines". I have looked into the American Psychological Association guidelines, and was shocked at how bad some of them are. To report two decimal places for everything, regardless of context, is simply wrong. Summaries for age, weight, waist circumference, blood pressure, minutes of activity, and EQ-5D VAS, should all be given to one decimal place. Given that the denominators are less than 100, percentages can be presented as whole numbers, or to one decimal place at most. The cost data could also be reported to the nearest AUS$.

Another area where the APA has given some rather odd advice is on the use of leading zeros. The authors stated in their response to my original comments, that this has been changed throughout, but it has not. The APA guildeline is to omit the leading zero if the quantity cannot exceed 1 (or -1). In my opinion, based on over 30 years studying and practising statistics, this is not good advice. Aesthetically, numbers between +/- 1 without the leading zero just look bad.

I still have reservations about the presentation of the EQ-5D data. In my experience, it is unusual to average the five items to produce a summary score for each individual. It is far more common to report the derived health utility score at each time point.

Minor point: in the section "What do these finding mean?", under the second point, I would add that studies with longer follow-up are needed.

Another minor point (which may be ignored) is that I would still say that the analysis was done according to the Intention To Treat principle, because individuals were (I assume) included in the analysis according to their randomised group allocation, regardless of their participation in the intervention. My earlier comment was to avoid connecting ITT with the issue of missing data. The new text to describe the analysis in this respect is good.

[LINK]

---

## [Editor Report · Decision Letter 2]

16 Jun 2020

Dear Dr. Quested, 

On behalf of my colleagues and the academic editor, Dr. Colin J Greaves, I am delighted to inform you that your manuscript entitled "A gender-sensitised weight loss and healthy living program for men with overweight and obesity in Australian Football League settings (Aussie-FIT): A pilot randomised controlled trial." (PMEDICINE-D-19-03524R2) has been accepted for publication in PLOS Medicine. 

PRODUCTION PROCESS

PRESS

PROFILE INFORMATION

Thank you again for submitting the manuscript to PLOS Medicine. We look forward to publishing it. 

Best wishes, 

Adya Misra, PhD

Senior Editor 

PLOS Medicine

plosmedicine.org